Genetic editing of the virulence gene of Escherichia coli using the CRISPR system

Hou Meijia
Sun Simeng
Feng Qizheng
Dong Xiumei
Zhang Ping
Shi Bo
Liu Jiali
Shi Dongfang shidf@neau.edu.cn
Veterinary Medicine, Northeast Agricultural University , Harbin , Heilongjiang Province , China
Souza Valeria
Electronic publication date: 2020 Apr 6
Publication date: 2020
Volume: 8
Electronic Location ID: e8881
Received 2019 Dec 2; Accepted 2020 Mar 10
Copyright: ©2020 Hou et al.
Copyright year: 2020
Copyright holder: Hou et al.
License: This is an open access article distributed under the terms of the Creative Commons Attribution License, which permits unrestricted use, distribution, reproduction and adaptation in any medium and for any purpose provided that it is properly attributed. For attribution, the original author(s), title, publication source (PeerJ) and either DOI or URL of the article must be cited.
License URL: https://creativecommons.org/licenses/by/4.0/

Keywords: CRISPR-Cas9, Virulence genes, ETEC, Single base mutation, CDS

Funding: National Science and Technology Support Project 2012BAD12B03-3 2012BAD12B05-2 Science and Technology Planning Project of Heilongjiang Province GC12B303 Financial support for this study was provided by grants from the National Science and Technology Support Project (2012BAD12B03-3, 2012BAD12B05-2) and the Science and Technology Planning Project of Heilongjiang Province (GC12B303). The funders had no role in study design, data collection and analysis, decision to publish, or preparation of the manuscript.

==============================
Clustered regularly interspaced short palindromic repeats (CRISPR)/Cas9 is an emerging gene-editing technology that is widely used in prokaryotes and eukaryotes. It can realize the specific manipulation of the genome efficiently and accurately. CRISPR/Cas9 coupled λ-Red recombination technology was used to perform genome editing in different genes. For finding an efficient method to edit the virulence genes of enterotoxigenic E. coli (ETEC), the two-plasmid system was used. The coding sequence (CDS) region of the estA, eltI, estB, eltIIc1, and faeG locus were deleted. The coding region of estB was substituted with estA. Gene recombination efficiency ranged from 0 to 77.78% when the length of the homology arm was from 50 to 300 bp. Within this range, the longer the homology arm, the higher the efficiency of genetic recombination. The results showed that this system can target virulence genes located in plasmids and on chromosomes of ETEC strains. A single base mutation was performed by two-step gene fragment replacement. This study lays the foundation for research on virulence factors and genetic engineering of vaccines for ETEC.

Introduction

Genome editing refers to the introduction of engineered mutations into specific loci of a target genome by genome editing techniques. In general, the current genome editing methods can be divided into three categories: the first category of genome editing relies on Cre (cause recombination enzyme)/loxP and Flp (Flippase recombination enzyme)/FRT (FLP recombination target); the second category is marker-free editing by the ‘pop-in/pop-out’ method; and the third category of genome editing relies on nucleases that recognize specific sequences.

Cre/loxP originates from the P1 phage. Cre is a recombinase that recognizes a specific genomic sequence called the loxP site. Cre recombinase removes sequences located between loxP sites (Abremski & Hoess, 1985). Flp/FRT originated from a plasmid in yeast. Flp is a recombinase similar to Cre, and its recognition site is named FRT (Hoang et al., 1998; Kilby, Snaith & Murray, 1993). The implementation of Cre/loxP and Flp/FRT technologies is generally divided into two parts. First, the marker gene with the screening function is integrated into the genome to complete the transformation of the target region, and then the resistance marker gene is excised by the recombinase. However, after each round of genetic manipulation, a loxP or FRT site is left on the genome, which is very unfavorable for the study of functional genes or multigene continuous operation.

To achieve seamless editing of the genome, the researchers chose a genetic framework with a positive screening function and a reverse screening function through two rounds of genetic manipulation. In addition, homologous recombination between the two rounds of genome editing required the expression of the exogenous λ-Red recombinase.

The CRISPR/Cas-mediated genome editing technology that has emerged in recent years is the third generation of endonuclease-dependent genome editing technologies. The principle is based on the mechanism of bacterial resistance to the invasion of foreign phage DNA (Sander & Joung, 2014). Compared with ZFNs and TALENs, CRISPR/Cas has the advantages of simple operation, a short operating cycle and low technical requirements for the operator. The CRISPR/Cas9 system has been successfully applied in diverse organisms including but not limited to humans (Drost et al., 2015), mice (Aida et al., 2015), zebrafish (Varshney et al., 2016), C. elegans (Shen et al., 2014), rice (Yin et al., 2017), E. coli (Jiang et al., 2013), and Saccharomyces cerevisiae (Brune, Kunze-Schumacher & Kolling, 2019), Chlamydomonas chloroplasts (Yoo et al., 2020).

For gene editing in E. coli, a three-plasmid system with highly efficient recombination was developed (Pyne et al., 2015). Subsequently, a two-plasmid system was constructed which included pCas containing the Cas9 protein and pTargetF containing the sgRNA (single guide RNA) (Jiang et al., 2015). However, the two-plasmid system still requires two steps in the process of curing the plasmid. To facilitate operation, a single-plasmid system was constructed which included Cas9 and sgRNA on one plasmid and it could be cured in one step by a temperature-sensitive element (Zhao et al., 2016). The CRISPR/Cas-mediated genome editing technology has been used in various articles for the purposes of deleting or replacing different virulence genes. However, we have not read any report on the editing of Escherichia coli virulence genes yet. At present, only genetic engineering model strains such as E. coli K12, DH5α and BL21 have been selected for CRISPR research. ETEC is one of the leading causes of bacterial diarrhea in humans and piglets. ETEC colonizes the intestines through the pili and then produces enterotoxin, which causes electrolyte disturbances that cause diarrhea. ETEC affects intestinal immunity through NF-κB and MAPK signaling pathways (Yang et al., 2016). Liu et al. (2015) studied several virulence genes of ETEC and constructed an attenuated strain with the λ-Red recombination system in our laboratory, but the recombination efficiency was low. Due to the complicated operation steps of three-plasmid system and the high construction cost of single-plasmid system, we selected the two-plasmid system for genetic editing in ETEC. The aim of this study was to find an efficient editing method for virulence genes located on chromosomes and in plasmids. This study paved the way for further research on virulence gene function, immune mechanism and genetically engineered vaccines of ETEC.

Material and Methods

Bacterial strains, plasmids, oligonucleotides, and growth conditions

The bacterial strains and plasmids used in this work are listed in Table 1. E. coli Trans1-T1 was used as a cloning host. The ETEC strains of E. coli O141:K85, E. coli O142 and E. coli DN1502 were used in the genome engineering after the susceptibility test, especially kanamycin and spectinomycin. E. coli O141:K85 (CVCC197) was a porcine ETEC field-isolated strain deposited in the Chinese Veterinary Culture Collection Center. E. coli O142 and E. coli DN1502 were isolated from diarrhea claves, and their virulence were tested in our laboratory (Liu et al., 2015; Yuan et al., 2014). Y-1 mouse adrenocortical cells were purchased from Shanghai Cell Bank, Chinese Academy of Sciences. Based on the gene sequence published in NCBI and the pTargetF plasmid sequence published in Addgene, primers were designed using Primer 5.0 software, and the N20 sequence was designed using online software. E. coli were grown in lysogeny broth (1% tryptone, 0.5% yeast extract, 1% NaCl) at 30 °C or 37 °C. Kanamycin (30 µg/mL), ampicillin (100 µg/mL), and spectinomycin (250 µg/mL) were added to the medium as appropriate. L-arabinose (1.5 mg/mL final concentration) was used for λ-Red induction when necessary.

Table 1 Strains and plasmids used in this studya.

Strains and plasmids	Characteristics	Source or reference	
E. coli Trans1-T1	F−ϕ80(lac Z) ΔM15 Δlac X74hsd R(rk−, mk+) Δrec A1398end A1ton A	TransGen Biotech	
E. coli O141:K85	elt I, estB, faeG	CVCC197	
E. coli DN1502	elt II c1	This laboratory (Liu et al., 2015)	
E. coli O142	estA	This laboratory (Yuan et al., 2014)	
pMD19T	bla	Taraka	
pCas	kan, Cas9, araC, Gam, Bet, Exo, repA101	Kindly provided by Pro. Yang Sheng	
pTargetF	sgRNA, aadA, pMB1	Kindly provided by Pro. Yang Sheng	
pTarget-Δelt I	aadA, pMB1, sgRNA-elt I, Δelt I (1,173 bp)	This study	
pTarget-ΔestB	aadA, pMB1, sgRNA-estB, ΔestB (220)	This study	
pTarget-Δelt II c1(400)	aadA, pMB1, sgRNA-elt II c1, Δelt II c1(400) (1,136)	This study	
pTarget-ΔestA	aadA, pMB1, sgRNA-estA, ΔestA (221)	This study	
pTarget-ΔfaeG	aadA, pMB1, sgRNA-faeG, ΔfaeG (474)	This study	
pTarget-ΔestB::estA	aadA, pMB1, sgRNA-estB, ΔestB (220)::estA (505)	This study	
pTarget-Δelt II c1p::faeG	aadA, pMB1, sgRNA-elt II c1, Δelt II c1p (477 bp)::faeG (234)	This study	
pTarget-ΔfaeG:: elt II c1193	aadA, pMB1, sgRNA-faeG, ΔfaeG (234):: elt II c1193(477)	This study	
pTarget-Δelt II c1(300)	aadA, pMB1, sgRNA-elt II c1, Δelt II c1(300) (1,136)	This study	
pTarget-Δelt II c1(200)	aadA, pMB1, sgRNA-elt II c1, Δelt II c1(200) (1,136)	This study	
pTarget-Δelt II c1(100)	aadA, pMB1, sgRNA-elt II c1, Δelt II c1(100) (1,136)	This study	
pTarget-Δelt II c1(50)	aadA, pMB1, sgRNA-elt II c1, Δelt II c1(50) (1,136)	This study	
Notes.

a bla, kan and aadA, represent resistance genes of ampicillin, kanamycin and spectinomycin, respectively; sgRNA-eltI, sgRNA with an N20 sequence for targeting eltI gene; Δelt I(1,173 bp), editing template for an 1,173-bp eltI deletion; ΔestB (220), editing template for a 220-bp estB deletion; Δelt II c1(400) (1,136), editing template for a 1,136-bp elt II c1 deletion with 400-bp homology arm; Δ estA(221), editing template for a 221-bp estA deletion; ΔfaeG (474), editing template for a 474-bp faeG deletion; ΔestB (220)::estA (505), editing template for a 220-bp estB deletion with a 505-bp estA insertion; Δelt II c1 (477 bp)::faeG (234), editing template for a 477-bp partial elt II c1 deletion with a 234-bp faeG insertion; ΔfaeG(234)::elt II c1193 (477), editing template for a 234-bp faeG deletion, with a 477-bp elt II c1193 insertion which contains the mutation of the 193rd amino acid of LT-II c1; Δelt II c1(300) (1,136), editing template for a 1,136-bp elt II c1 deletion with 300-bp homology arm; Δelt II c1(200) (1,136), editing template for a 1,136-bp elt II c1 deletion with 200-bp homology arm; Δ elt II c1(100) (1,136), editing template for a 1,136-bp elt II c1 deletion with 100-bp homology arm; Δelt II c1(50) (1,136), editing template for a 1,136-bp elt II c1 deletion with 50-bp homology arm. The length of the homology arm in this article refers to the length of one side. Donor primers were used to amplify homologous arms.

Plasmid construction

We used the plasmid constructed by Sheng Yang ’s laboratory (Jiang et al., 2015) to edit the virulence gene of ETEC. These virulence genes included type II heat-labile enterotoxin c1 subtype LT-IIc1 gene (elt II c1) located on the chromosome; type I heat-labile enterotoxin LT-I gene (elt I), type I heat-stable enterotoxin gene STa (estA) and type II heat-stable enterotoxin STb gene (estB) located on the pEnt plasmid; and FaeG subunit gene (faeG) of K88 fimbriae located on other plasmids. First, the homology arms were amplified using the primers pairs DonorL-F/R and DonorR-F/R to form DonorL and DonorR. 2 µL bacterial solution were used as DNA sample and subjected to PCR reactions where each reaction in 25 µL consisted of 16 µL of deionized water, 1 µL of forward primer, 1 µL of reverse primer, 2 µL 2.5mM dNTPs, 2.5 µL 6X DNA Buffer and 0.5 µL EasyTaq DNA Polymerase (TransGen, 126 Beijing, China). The PCR amplification was performed as follows: initial denaturation, 94 °C for 5 min; 30 cycles of 94 °C for 30 s, 55 °C for 30 s and 72 °C for 45 s; final polymerization 72 °C for 7 min. The PCR products were separated on 0.8% agarose gel by electrophoresis. The DNA bands were cut out of the gel and extracted using the Gel Extraction Kit (Omega, Norcross, GA, USA). Then, N20 and sgRNA fragments were amplified from pTargetF by the primer pair psgRNA-F/R. The targeted N20 was designed according to GenBank ID with the help of a website (https://zlab.bio/guide-design-resources).

The homologous arms with N20 and sgRNA fragments were ligated into the pMD19T (Taraka, Dalian, China) vector after overlap PCR. The ligation system contained 1 µL of pMD19T, 3 µL of DNA fragments, 5 µL of Solution I, and 1 µL of deionized water. Mixed the above reagents and left at 16 °C for 1 h. The ligation product was introduced into DH5α. After sequencing, the correct recombinant pMD19T was named as repMD19T. repMD19T and pTargetF were digested with Spe I and Xho I, and then the digested small fragment of repMD19T and large fragment of pTargetF were ligated overnight at 16 °C using T4 ligase. In this way, pTarget-Δelt I, pTarget-ΔestB, pTarget-Δelt II c1(400), pTarget-ΔestA and pTarget-ΔfaeG were constructed. pTarget-ΔestB::estA was constructed by inserting the overlap PCR fragment into Spe I/Xho I-digested pTargetF, which was amplified with primers P9/P10, P11/P41, P42/P43, and P44/P14. pTarget-Δelt II c1 p::faeG was constructed by inserting the overlap PCR fragment into Spe I/Xho I-digested pTargetF, which was amplified with primers P17/P45, P46/P47, P48/P49, and P50/P51. pTarget-ΔfaeG::elt II c1193 was constructed by inserting the overlap PCR fragment into Spe I/Xho I-digested pTargetF, which was amplified with primers P33/P45, P46/P52, and P53/P51. pTarget- Δelt II c1(300), pTarget-Δelt II c1(200), pTarget-Δelt II c1(100), and pTarget-Δelt II c1(50) was constructed by inserting various lengths of homology arm amplified with pTarget-Δelt II c1 as a template into Xba I/Xho I-digested pTarget-Δelt II c1, respectively.

Genome editing procedure

ETEC was used in all mutagenesis experiments. The electro competent cells of original bacterial were prepared as follow steps. One single colony of ETEC from an LB agar plate was cultured overnight in 3 mL of LB medium at 37 °C. 1 mL of overnight culture was inoculated and grown in 100 mL of LB medium at 37 °C until the optical density at 600 nm (OD600) reached 0.6∼0.8. Then, the cells were centrifuged at 5,000 rpm for 15 min at 4 °C and washed three times with ice-cold 10% glycerol. After the final wash, 1 mL of ice-cold 10% glycerol was added to the tube to resuspend the cells, which were dispensed into a frozen EP tube per 100 µL. pCas was introduced into the competent cells. Transformation was done by electroporation using the ECM 399 Electroporation System (BTX). Electro competent cells harboring pCas were prepared by growing a single colony in 3 mL of LB medium with 30 µg/mL kanamycin at 30 °C. One milliliter of overnight culture was inoculated and grown in 100 mL of LB medium at 30 °C. L-arabinose was added to a concentration of 1.5 mg/mL at 0.2∼0.3 of OD600. Then, the cells were harvested at an OD600 of 0.6∼0.8 by centrifugation, washed three times with ice-cold 10% glycerol, and concentrated approximately 100-fold. 100 µL of electro competent cells were mixed with 300 ng of pTarget series plasmids. Electroporation was performed in a 2-mm gap electroporation cuvette (BTX) at 2.5 kV. The electro transformed cells were suspended immediately in 1 mL of LB medium and recovered at 30 °C for 1 h, after which they were plated on LB agar containing kanamycin and spectinomycin and incubated overnight at 30 °C. A single colony was picked and inoculated into LB medium containing kanamycin and spectinomycin for the next tests. Transformants were identified by PCR and DNA sequencing.

Plasmid curing

For pTarget curing, the identified bacteria were streaked onto LB plates containing kanamycin and spectinomycin and cultured overnight at 30 °C. Then, a single colony was picked and inoculated into 3 mL of LB medium containing kanamycin and 1 mM isopropyl-β-D-thiogalactoside (IPTG) and cultured at 30 °C for 16 h. The culture was confirmed as cured by ensuring the sensitivity to spectinomycin (250 µg/mL) and by PCR method. For pCas curing, the identified bacteria were streaked on an LB plate containing kanamycin and cultured at 30 °C overnight. Then, a single colony was picked and placed in 3 mL nonselective LB medium and cultured at 37 °C for 16 h. The cultures were confirmed as cured by measuring their sensitivity to kanamycin (30 µg/mL) and PCR method. The gene editing theory and process are shown in Figs. 1 and 2, respectively.

Figure 1 Gene editing theory.

(A) First, the host strain to be mutagenized is transformed with pCas expressing the λ Red component (Exo, Bet, Gam), the Cas9 endonuclease, and tracrRNA. Next, the host strain containing pCas is transformed with pTarget series carrying donor DNA and encoding the gRNA that specifies the site of cleavage. The gRNA directs the Cas9 endonuclease to the cleavage site. (B) While the gRNA recognizes 20 bases of the target site, the Cas9 mediates bacterial DNA double strand break (DSB). DSB is repaired by λ Red homologous recombination.

Figure 2 Gene editing and plasmid curing process.

Step I: pCas was introduced into the host cell. Step II: pTarget was introduced into the host cells harboring pCas. Step III: Targeted gene of the host cells was recombined. Step IV: pTarget was cured with IPTG and cultured recombined cells at 30 °C. Step V: pCas was cured by cultured recombined cells at 37 °C.

Results

Editing of the ETEC virulence gene

We constructed single-gene deficient and multi-genes deficient ETEC strains. For a single gene deletion, 10% of the transformants showed the expected genotype when elt I was deleted from E. coli O141:K85 (Table 2, experiment 1). The efficiency of knocking out estB and faeG from E. coli O141:K85 was 1.4% and 23.8%, respectively. We tested the continuous virulence gene deletion ability of this two-plasmid system in ETEC. Three virulence genes of E. coli O141:K85 were deleted with pTarget-Δelt I, pTarget-ΔestB, pTarget- ΔfaeG. Then, E. coli O141:K85 Δelt I, E. coli O141:K85 Δelt I Δestb, and E. coli O141:K85 Δelt I ΔestB ΔfaeG were obtained (Table 2, experiments 1, 3, 5). These results proved that the system can continuously edit the virulence genes of ETEC.

Table 2 Mutation efficiency of pCas/pTarget system.

Expt. no.	Targeting genome locus of sgRNA	Host cell	Plasmid pTarget	Length of homology arm (left, right) (bp)	Number of picked colonies /recombinant colonies	Mutation efficiency (%)	
1	elt I	E. coli O141:K85	pTarget-Δelt I	137,132	50/2	4.00	
2	estB	E. coli O141:K85	pTarget-ΔestB	167,226	68/1	1.40	
3	estB	E. coli O141:K85 Δelt I	pTarget-ΔestB	167,226	85/15	17.64	
4	faeG	E. coli O141:K85	pTarget-ΔfaeG	203,202	21/5	23.80	
5	faeG	E. coli O141:K85 Δelt IΔestB	pTarget-ΔfaeG	203,202	50/2	4.00	
6	estB	E. coli O141:K85	pTarget-ΔestB::estA	167,226	99/14	14.14	
7	estA	E. coli O142	pTarget-ΔestA	112,101	115/9	7.80	
8	elt II c1	E. coli DN1502	pTarget-Δelt II c1(400)	404,465	75/51	68	
9	elt II c1	E. coli DN1502	pTarget-Δelt II c1p::faeG	309,428	40/36	90	
10	faeG	E. coli DN1502 Δelt II c1::faeG	pTarget-ΔfaeG:: elt II c1193	309,428	20/20	100	
11	elt II c1	E. coli DN1502	pTarget-Δelt II c1(300)	304,331	42/54	77.78	
12	elt II c1	E. coli DN1502	pTarget-Δelt II c1(200)	235,202	26/43	60.47	
13	elt II c1	E. coli DN1502	pTarget-Δelt II c1(100)	124,103	1/60	1.67	
14	elt II c1	E. coli DN1502	pTarget-Δelt II c1(50)	49,49	0/19	0	
Notes.

Experiments 1, elt I was deleted from E. coli O141:K85. Experiments 2, estB was deleted from E. coli O141:K85. Experiments 3, estB was deleted from O141:K85 Δelt I. Experiments 4, faeG was deleted from E. coli O141:K85. Experiments 5, faeG was deleted from E. coli O141:K85 Δelt IΔestB. Experiments 6, estA was replaced in estB loci in E. coli O141:K85. Experiments 7, estA was deleted from E. coli O142. Experiments 8, elt II c1 was deleted from DN1502 with 400 bp homology arm. Experiments 9–10, the codon of the 193rd amino acid (leucine) of elt II c1 was changed from CTG to CTC. Experiments 11–14, elt II c1 was deleted from DN1502 with different length of homology arms.

Comparing the deletion of different virulence genes, we found that the mutation rate was lower than 25% when the length of the homology arm fragment was less than 200 bp (Table 2, experiments 1, 2, 4). But the mutation rate increased significantly to more than 60% when the length of the homology arm was over 300 bp (Table 2, experiments 6–10). Besides gene deletion, we also carried out gene replacement. The mutation efficiency was 14.14% to replace estB with estA (Table 2, experiment 6).

For point mutations, pTarget-Δelt II c1::faeG was transformed into E. coli DN1502 harboring pCas to replace elt II c1 with faeG to obtain E. coli DN1502 Δelt II c1::faeG. We constructed pTarget-ΔfaeG::elt II c1193 which sgRNA targeted faeG and contained a mutant elt II c1 fragment. Then, pTarget-ΔfaeG::elt II c1193 was electrotransformed into the bacteria obtained in the previous step. This changed the 193rd amino acid codon of elt II c1 from CTG to CTC, forming a Sac I cleavage site (GAG192CTC193). These results proved that continuous gene editing and point mutations can be achieved (Table 2, experiments 9, 10).

Since the efficiency of gene deletion ranged, there is necessary to further explore whether the size of the homologous arm was related to it. We designed the pTarget-Δelt II c1 series with 300 bp, 200 bp, 100 bp, 50 bp homologous arms, combined with pTarget-Δelt II c1(400) (Table 2, experiments 8, 11–14). By comparison, the gene editing efficiency was the highest when the homology arm was approximately 300 bp. The homology arms size, deletion fragment length and recombination efficiency of the target genes are listed in Table 2. Agarose gel electrophoresis of colony PCR is shown in Fig. 3.

Figure 3 Agarose gel electrophoresis of colony PCR for different genes and SacI digestion.

(A) Identification of eltI gene knockout with primers P7/P8. Lane 1, positive control. Lane 2, Negative control. (B) Identification of estB gene knockout with primers P15/P16. Lane 1, positive control. Lane 2, Negative control. (C) Identification of faeG gene knockout with primers P39/P40. Lane 1, positive control. Lane 2, Negative control. (D) Identification of eltIIc1 gene knockout with primers P23/P24. Lane 1, positive control. (E) Identification of estA inserted into estB with primers P15/P16. Lane 1, positive control. Lane 2, Negative control. (F) Identification of estA gene knockout with primers P31/P32. Lane 1, positive control. Lane 2, Negative control. (G) Identification of faeG inserted into eltIIc1 with primers P54/P55. Lane 1, positive control. Lane 2, Negative control. (H) Identification of eltIIc1193 gene single base mutation with primers P54/P55. Lane 1, positive control. Lane 2, Negative control. (I) Lane 1, eltIIc1193 of the corrected recombinant ETEC was amplified with primers P23/P24. Lane 2, amplified fragment was digested with SacI for 3 h.

Plasmid curing

For the pTarget series plasmid curing, the effects of IPTG concentration and incubation time were tested. IPTG (0.5 mM and 1 mM) was used to induce the Ptrc promoter to guide sgRNA to target the pTarget series plasmid. Then, cells were cultured at 30 °C for 8, 10, 12, 14 and 16 h. The results showed that 1 mM IPTG induction with 8 h culture made the optimal combination. The colonies on the kanamycin-containing plates grew normally, while there was no visible growth on the spectinomycin-containing plates. The pTarget series plasmids were verified to be eliminated by colony PCR with primers P68/P69. pTarget series plasmids could be cured in all of the genetically edited strains.

For the pCas plasmid curing, incubation times were tested for 8, 10, 12, 14, and 16 h. The results showed that 8 h for curing pCas was enough. The colonies grew normally on the LB plates, and there was no visible growth on the kanamycin plates, indicating that the elimination of pCas was successful. pCas could be cured in all of the genetically edited strains. Agarose gel electrophoresis of colony PCR with primers P70/P71 after plasmid curing is shown in Fig. 4.

Figure 4 Agarose gel electrophoresis of colony PCR after curing the plasmids.

(A) Verification of eltI gene knockout and plasmid curing. (B) Verification of eltIIc1 gene knockout and plasmid curing. (C) Verification of estB gene knockout and plasmid curing. (D) Verification of estA inserted into estB and plasmid curing. (E) Verification of faeG inserted into eltIIc1 and plasmid curing. (F) Verification of eltIIc1193 gene single base mutation and plasmid curing. (G) Verification of faeG gene knockout and plasmid curing. (H) Verification of estA gene knockout and plasmid curing. Region 1: Verification of editing fragments after curing the plasmids. Region 2: Verification of pTarget series after curing with IPTG, using primers P68/69. Region 3: Verification of pCas after curing with temperature sensitive replicon, using primers P70/P71. The first three lanes in each region are genetically edited ETEC. The fourth is a positive control and the fifth is a negative control.

Discussion

The CRISPR/Cas gene operating system has made great progress in the field of eukaryotes, but it is not widely used in prokaryotes. The two-plasmid system combined the CRISPR/Cas9 system and the λ-Red recombination to achieve target editing in the E. coli genome (Jiang et al., 2015), and the recombination efficiency was greatly improved. In this system, the temperature-sensitive plasmid pCas expressed Cas9, Exo, Bet, and Gam proteins. The last three proteins mediated homologous recombination in the λ-Red system. pTargetF expressed an sgRNA sequence that specifically recognized the target site. There is an IPTG-induced transcription of sgRNA on pCas, which is directed to the replicon pMB1 of pTargetF. Therefore, cas9+sgRNA cleaves pMB1 to cleave and eliminate pTargetF.

To date, model strains such as E. coli K12 and E. coli BL21 (DE3) have been employed when using CRISPR/Cas in E. coli for genetic manipulation. However, there have been no reports of genetic manipulation in wild-type E. coli. In this study, we focus on the application and efficiency of the CRISPR/Cas system in wild-type ETEC virulence genes.

This two-plasmid system introduces Red recombination technology into CRISPR. The Gam protein can inhibit the RecBCD exonuclease of the host cell from degrading exogenous linear DNA. That made linear fragment could be used as the donor DNA. In fact, we tried to use linear fragment as the donor DNA early in the experiment. But there were no expected results. In Jiang’s article, linear fragment donors were less efficient than circular plasmid donors (Jiang et al., 2015). This is consistent with the results of our actual operation. Thus, donor DNA was assembled into pTarget in this study.

We tested whether continuous gene knockout can be performed in wild-type E. coli. After deleting single gene elt I, estB, and faeG from the original E. coli O141:K85 respectively, we successfully deleted the three genes consecutively in the bacteria. The deletion efficiency did not change significantly. This indicated that the two-plasmid system is efficient tool for editing virulence genes of ETEC in the laboratory.

We also achieved a single base mutation in elt II c1 in E. coli DN1502 by two rounds of substitutions in the same position. This changed the codon of the 193rd amino acid (leucine) of elt II c1 from CTG to CTC where a Sac I restriction enzyme cutting site emerged. This operation has not been reported before our study. It proved the feasibility of continue substitution at the same position in ETEC, and improved the application of the two-plasmid system.

Because we wanted to delete the complete CDS region of the virulence gene, the size of the deletion fragment depended on the target gene itself. Some published sequences have only the CDS region or are only slightly longer than the CDS region. The size of homology arm was affected by sequence published in GenBank (Table 3). To explore the effect of the length of the homology arm on the recombination efficiency, we designed 5 plasmids to knock out elt II c1. It was found that the longer the homology arm, the higher editing efficiency when the deletion fragment length was the same. The deletion efficiency was the highest when the length of the homology arm was approximately 300 bp. The toxicity of E. coli was significantly reduced after the virulence gene was deleted. Relevant experiments and results were shown in Supplementary Materials. Berger et al. (2016) analyzed the pAA primary transcriptome using differential RNA sequencing and provided novel insights into its virulence gene expression and regulation. A recent study showed that in patients with lung adenocarcinoma, patients with lymph node metastasis had significantly elevated expression of long non-coding RNA than patients with non-lymph node metastases. This indicated that long noncoding RNAs could be seen as candidate diagnostic and prognostic biomarkers for lung adenocarcinoma (Wang et al., 2019). Does the non-coding region affect the toxicity of ETEC? Is the virulence gene transcribed if the promoter is deleted? These questions are interesting and worthy of further exploration.

Table 3 N20 + PAM sequence and GenBank ID.

Target gene	N20+PAM	GenBank	
elt I	AAGCTTGGAGAGAAGAACCC TGG	CP002732.1	
elt II c1	TCTTTTGGTGCGATA GAAGG GGG	JQ031705.1	
estB	CAAATAATGGTTGCAGCAAA AGG	AY028790.1	
faeG	GCCGGTGTGTTCGGGAAAGG TGG	V00292.1	
estA	TGTTGTAATCCTGCCTGTGC TGG	V00612.1	
Notes.

a Primers for virulence genes and N20 are designed according to the GenBank ID in the table.

The elt I, estA, estB and faeG locous are located on the plasmid of E. coli, and the elt II c1 loci is located on the E. coli chromosome. It demonstrated that the two-plasmid-based CRISPR-Cas9 system can target genes in plasmids and chromosomes, which was important to edit virulence genes in ETEC.

To identify the recombinant strain by PCR, the pair of primers on the outside of the homology arm was required. If homologous arm primers were used, it would result in false positives. Because the pTarget plasmid with the homologous arm was not eliminated at this time.

The specificity of the CRISPR/Cas9 system depends mainly on sgRNA (Alkan et al., 2018; Fu et al., 2014). The designed sgRNA may form a mismatch with nontarget DNA sequences, resulting in unintended gene mutations, that are called off-target effects (Zhang et al., 2015). The off-target mutations can cause genomic instability and disrupt the function of other normal genes (Hsu et al., 2013; Mali et al., 2013). The off-target effect of Cas9 has been reported frequently in eukaryotes (Fu et al., 2013; Hsu et al., 2013). When we knocked out the estB, the first selected colony of N20 did not complete the knockout task. Another colony of N20 was replaced to continue and then succeeded. To reduce the off-target effects of Cas9 in this study, it was best to ensure that the 12 bases at the 3′end of the N20 sequence were highly specific to the genome (Jiang et al., 2013; Jiang et al., 2015). Existing online sgRNA design tools are not specifically targeted at ETEC. The evaluation of sgRNA off-target probability is based on E. coli K12 MG1655 instead of ETEC. It might lead to inaccurate evaluation of sgRNA off-target probability. From this current experimental results, the knockout efficiency can meet our requirements for gene editing.

The two-plasmid CRISPR/Cas system gives us more convenient conditions to study ETEC. The two-plasmid CRISPR/Cas system gives us more convenient conditions to study ETEC. It could be used to explore amino acids that affected virulence through site-directed mutations. And it also could be used to display foreign proteins on outer membrane or fimbriae of ETEC to stimulate the body to produce antibodies. We believe that the two-plasmid system will be more widely adopted following the further research.

Conclusions

ETEC is one of the important pathogens causing human and animal diarrhae. But the researches of its virulence genes were affected due to the lack of efficient gene editing tools. In this study, we explored the validity of the two-plasmid system of CRISPR/Cas for editing the virulence genes in ETEC. The results show that the two-plasmid system of CRISPR/Cas used in this study can edit the virulence genes on the ETEC plasmids and chromosome for deleting, replacing, and mutating. The editing efficiency is closely related to the homologous arm length, and the efficiency is the highest when the length of the homology arm is approximately 300 bp. This system can also be used for single base mutation through two rounds of replacement at the same position. These studies about how to quickly and efficiently perform gene editing in ETEC using the two-plasmid-based CRISPR-Cas9 system lay the foundation for further research on the roles of virulence genes in pathogenicity, antigenicity and adjuvanticity, and mutation of virulence genes for genetically engineering a vaccine of ETEC.

Supplemental Information

Table S1 Primer sequences

Underline is the enzyme cleavage site, the italic is N20, and the double underline is the part of the pTarget plasmid. *Primers are from http://www.addgene.org/.

Click here for additional data file.

Figure S1 Sequencing results of recombinant genes

Click here for additional data file.

Figure S2 Plasmids digested with SpeI and XhoI

From left to right are: pTarget, pTarget-Delt, pTarget-Δstb, pTarget-stb::estA, pTarget-ΔfaeG, pTarget-ΔestA, pTarget-ΔeltIIc1, pTarget-ΔeltIIc1::faeG, pTarget-ΔfaeG:: eltIIc1193

Click here for additional data file.

Figure S3 Toxic effects of ETEC enterotoxins

Click here for additional data file.

Supplemental Information 1 Nucleic acid electrophoresis scans

Click here for additional data file.

The authors are grateful to Pro. Sheng Yang (Key Laboratory of Synthetic Biology, Institute of Plant Physiology and Ecology, SIBS, CAS, Shanghai, China) for his kind provision of pCas and pTargetF and his patient guidance. His protocol has given us much help, and we have had great convenience in experimentation. The plasmids pCas and pTargetF described in his work have been deposited in Addgene (http://www.addgene.org/) under No. 62225 and No. 62226, respectively. We also thank Pro. Yudong Cui for providing E. coli O142.

Additional Information and Declarations

Competing Interests

Author Contributions

Data Availability

The authors declare there are no competing interests.

Meijia Hou conceived and designed the experiments, performed the experiments, prepared figures and/or tables, and approved the final draft.

Simeng Sun performed the experiments, prepared figures and/or tables, and approved the final draft.

Qizheng Feng analyzed the data, prepared figures and/or tables, and approved the final draft.

Xiumei Dong and Ping Zhang analyzed the data, authored or reviewed drafts of the paper, and approved the final draft.

Bo Shi and Jiali Liu analyzed the data, prepared figures and/or tables, and approved the final draft.

Dongfang Shi conceived and designed the experiments, authored or reviewed drafts of the paper, and approved the final draft.

The following information was supplied regarding data availability:

The original data is available in the Supplemental Files.

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
