# Peer review of "Genetic editing of the virulence gene of Escherichia coli using the CRISPR system"

_PeerJ, doi:10.7717/peerj.8881_

## Round 0.1 · original submission · Minor Revisions

Although two reviewers found that the study needs minor revisions, one founds that it should be edited from professional English service.
The structure of the manuscript should be improved. Please rearrange your results.

Reviewer 1 ·

Basic reporting

good

Experimental design

good

Validity of the findings

good

Additional comments

Authors have used a two-plasmid system of CRISPR for the Genetic editing of the virulence gene of Escherichia coli and can be used for deleting, replacing, or mutating the ETEC associated virulence genes on plasmids and chromosome. This is a good mechanistic study with clear objectives. However I suggest some minor comments
1. Despite the fact that authors have mentioned ‘the editing efficiency is closely related to the homologous arm length, and the efficiency is the highest when the length of the homology arm is approximately 300 bp’. This high efficiency means to edit/replace coding sequence? Have you checked any non-coding DNA too?
2. Another important change should be made in the statement of the objectives. Actually, there is no such statement of the objectives of the study; in case there is one, it failed to catch my attention as a reader. In your revision, include, in a sentence or two, a clear statement of the objectives of the study. Because this technology is used in various articles for different deleting, or replacing purposes of virulence genes. What is the impact of your study?
3. You have examined the ETEC associated virulence genes editing. Or have you checked any clinical virulent strain? And did you check any lethality of the test strains? If so please include the lethality test results, as VAGs are not considered virulent based on molecular data only.

·

Basic reporting

Although the study is interesting the structure and the grammar are ambiguous. It should be edited from professional English service.
The structure of the manuscript should be improved.
Please rearrange your results.

Experimental design

The objectives are not clear.
Experimental design should be improved.
Methods are not sufficiently described.

Validity of the findings

Impact and novelty not assessed.
Conclusions are not stating the research question & limited to supporting results. It should be improved.

Additional comments

1. The authors mentioned that a two plasmid system (pCas/pTargetF) that contains Cas9 and λ-Red recombination elements constructed by Jiang. Who is Jiang? I think it should not be mentioned as the name in the abstract. You can write it in the method section.
2. The introduction is inconsistent. The paragraphs line 72 to 79 are unnecessary. Should be removed.
3. The authors mentioned that the E. coli O142 and E. coli DN1502 were isolated from the bovine rectum. How they isolated from the rectum directly? It is not logical. Please make it clear.
4. Please re-write the lines 124-128.
5. The authors mentioned experiment numbers throughout the manuscript. I could not understand about these experiments. Which experiments have been numbered in this manuscript?
6. There repetition of plasmid curing methods in the results. Please remove this paragraph from the results .
7. The results are few but the discussion section is too long and non-logical. The discussion must be re-written and it should be in a logical flow that compares your results.
8. The English language are poor. It should be edited by a professional or native English speaker.

Reviewer 3 ·

Basic reporting

Figure 2 requires detailed annotation;The electrophoretic bands in Fig. 3 and Fig. 4 need to be marked clearly with maker size;The annotation of the figure should be below the figure

Experimental design

no comment

Validity of the findings

no comment

Additional comments

The study seems to be quite interesting and is quite valuable to report on CRISPR/Cas9 coupled λ-Red recombination technology, which was firstly invented by Liu et al. As far, model strains such as E. coli K12 or E. coli BL21 have been employed when using CRISPR/Cas for genetic manipulation in E. coli, but nobody has performed genetic manipulation in virulence genes of wild-type Escherichia coli. In this study, the authors focus on the application and efficiency of the CRISPR/Cas system in enterotoxigenic E. coli (ETEC) virulence genes. The improved gene editing technology provides great convenience for the study of virulence genes of wild-type Escherichia coli. The manuscript is suitable for publication in “PeerJ” but in order to improve the quality of the text, following “minor changes” are suggested:
1. in line 44-45, How to understand this sentence “Gene recombination efficiency ranged from 0 to 77.78% when the length of the homology arm was from 50 to 300 bp” Is the efficiency of gene editing high or low?
2. in line 96-98, It is mentioned that “our laboratory selected the two-plasmid system constructed by Jiang to perform genetic editing in E.coli”, both are gene editing technology in these two article, what is the main innovation of this article.

---

## Round 0.2 · accepted · Accept

This version of the manuscript greatly improved both the language and the structure of the manuscript. All the reviewers concerns were met.